# Engagement in types of activities and frequency of alcohol use in a national sample of United States adolescents

**Cassandra A. Sutton[1,2], Elizabeth Grandfield[3], Richard Yi ◉[1,2], Tera L. Fazzino ◉[1,2]***

**1** Department of Psychology, University of Kansas, Lawrence, Kansas, United States of America, **2** Cofrin Logan Center for Addiction Research and Treatment, University of Kansas, Lawrence, Kansas, United States of America, **3** Department of Methodology and Statistics, Utrecht University, Utrecht, Netherlands

* tfazzino@ku.edu

## Abstract

### Objective

Adolescents with fewer sources of environmental reinforcement may be at risk for alcohol use. Behavioral economic theories posit that engagement in some activities may facilitate alcohol use, whereas other activities may be incompatible with use and reduce likelihood of alcohol use. It is unclear which types of activities may facilitate or may be incompatible with alcohol use in adolescence. Using a national sample of adolescents, the current study examined differences in engagement with types of activities that may be incompatible with alcohol use, compared among adolescents who endorsed alcohol use, and adolescents who did not.

### Method

Data from the 2019 Monitoring the Future (MTF) study ($N$ = 4626) were analyzed. Potentially incompatible and facilitating activities, and alcohol-involved activities were identified from pre-existing survey measures. Confirmatory factor analysis, measurement invariance, and structural equation modeling were used to examine patterns in activity engagement among those who endorsed alcohol use and those who did not.

### Results

Participants who did not endorse alcohol use reported higher engagement in activities that may be incompatible with alcohol use, including enjoyment from school and going to the mall ($p$ < .001). Participants who endorsed alcohol use reported higher engagement in activities that may facilitate alcohol use (p < .001), such as spending time with friends and attending parties. Facilitating activities ($\beta$ = 0.15, $p$ < .001) and alcohol-involved activities ($\beta$ = 0.70, $p$ < .001) were positively associated with alcohol use frequency. Observed effect sizes were small in magnitude for all findings.

**Data Availability Statement:** The data reported in the paper were obtained from publicly available data from the Institute for Social Research at the University of Michigan. Specifically, data were

obtained from the 8th- and 10th-grade sample of the 2019 Monitoring the Future study (ICSPR 37842). All data files can be accessed at the following location: https://www.icpsr.umich.edu/web/ICPSR/studies/37842.

**Funding:** A grant funded by the National Institute on Alcohol Abuse and Alcoholism (R01 AA027791-01; PI: Fazzino) supported some of CS, RY, and TLF's time during the study. The funders had no role in study design, data collection and analysis, decision to publish, or preparation of the manuscript.

**Competing interests:** The authors have declared that no competing interests exist.

## Conclusions

The findings support the premise of behavioral economic theory, suggesting some activities may serve as protective factors against alcohol use frequency while other activities may facilitate alcohol use among adolescents. National surveys may consider adding specific measure of activity engagement to identify activities that may be incompatible with alcohol use among adolescents.

## Introduction

Adolescence is a sensitive developmental period that is high risk for alcohol use [1]. During adolescence, individuals gain greater independence and responsibility for making choices regarding their daily activities (e.g., how to spend their free time, what to eat, etc.), while experiencing ongoing neurodevelopment in key brain regions that govern self-regulation and executive functioning [2,3]. These factors, combined with expanding access to alcohol in the environment, heighten the risk of adolescents engaging in alcohol use [1,4]. Consequences of alcohol use during adolescence may be substantial and can include impairments in learning processes, executive functioning, and decision-making [5]. Alcohol use in mid-adolescence (~16 years old) in particular has been associated with greater negative consequences from alcohol use that may span into early adulthood, relative to adolescents who initiate alcohol use later [6–9]. Thus, research is needed to understand circumstances that may facilitate alcohol use among those in mid-adolescence.

According to principles of behavioral economics, adolescents with greater environmental opportunities to use alcohol, and/or adolescents who have fewer alternative activities to alcohol use, may be more likely to use alcohol [10–13]. Opportunities to use alcohol in an adolescents' environment may come directly from situations in which alcohol is present (e.g., a party in which drinking is the central activity), as well as situations in which alcohol could be used to enhance adolescents' experiences, such as recreational activities without adult supervision (e.g., hanging out with friends). Thus, some situations and activities may facilitate use or leave the opportunity open for use [10,12,14]. In contrast, other activities may reduce the likelihood of use, or may be directly incompatible with use [10,12]. For example, engaging in moderate to vigorous physical activity may be considered incompatible with alcohol use, as drinking may be physically infeasible when exercising vigorously. Additionally, some circumstances may be incompatible with alcohol use over a longer period of time, such as doing well in or enjoying school. Thus, some types of activities may facilitate alcohol use and may serve to increase alcohol use frequency, whereas other activities may be incompatible with alcohol use, thereby reducing likelihood of alcohol use in adolescence.

Across studies examining activity engagement and substance use among adolescents, results have generally supported the premise that greater engagement in activities that may facilitate use may yield greater substance use, while greater engagement in alternative activities that are incompatible with use may yield lower substance use. Overall, findings from several large, regionally representative studies among high school students indicated that engagement in activities that facilitate substance use (e.g., spending unsupervised time with peers, attending parties, etc.) was positively associated with substance use cross-sectionally and predicted substance use longitudinally [10,11,15–18]. Findings have also indicated that reduced engagement in alternative activities may increase an adolescent's risk for engaging in substance use [18]. Findings regarding activities that may facilitate substance use or may be incompatible with

substance have been observed using multiple behavioral economic measures, including measures of activity engagement, measures of reinforcement from substance-related and substance-free activities, and measures of time allocation as a proxy measure of activity valuation and engagement [10,11,15–18]. Thus, evidence from several large, regionally representative samples of US high school students indicated that lower engagement with alternative activities, as well as greater engagement with activities that may facilitate use, may yield greater substance use behavior.

Despite the relatively strong evidence in the literature regarding the role of facilitating and incompatible activity engagement as factors in substance use, the types of activities that may reduce the likelihood of substance use among adolescents is largely unclear. Thus far, three regional studies have evaluated types of activities that may be incompatible with or limit engagement with alcohol use in adolescence, with varying degrees of examination. Across the studies, all identified family activities or family involvement as being negatively correlated with risky drinking, with small to moderate effect sizes [11,19,20]. In addition, one study that used regional data from the early to mid 1990's examined a variety of activities that may be incompatible with alcohol use, which included school-related activities, extracurricular activities, sports, time spent alone, paid work, and time spent on media (i.e., television, paper magazines); however, none of the activities were significantly associated with drinking [19]. Thus, it remains largely unclear whether certain types of activities may be incompatible with and may serve to limit engagement in alcohol use beyond involvement with family during adolescence. Furthermore, given the substantial societal changes observed over the past 30 years that may influence adolescents' engagement in activities (e.g., role of parental supervision, emergence of social media) and alcohol use, an updated analyses of the role of activity engagement and adolescent alcohol use is needed.

Behavioral economic theory suggests that increasing an adolescent's engagement with alternative activities to alcohol use may be an approach to reduce drinking. However, our understanding regarding activities that may be incompatible with or limit alcohol use in adolescence is very limited and provides prevention researchers little information with which to guide and tailor prevention approaches. In this regard, no nationally representative studies with adolescents have included established survey measures to characterize engagement with alternative or incompatible activities to substance use; thus, data regarding types of activities that may limit engagement with alcohol use are not available to the general research community. One potential alternative approach that could leverage existing data sources would be to use nationally representative survey data, collate survey items from pre-existing measures that address types of potentially incompatible activities (e.g., school activities, volunteering in the community, sports/physical activity, etc.), and test associations with alcohol use among adolescents. The approach would facilitate the use of available data from a nationally representative sample of adolescents to examine various types of activities that may serve to limit engagement with alcohol use. Furthermore, the approach would provide an opportunity to test whether the relationship between engagement in facilitating activities and alcohol use reported in studies with regional samples conducted 10–20 years prior may partially replicate using a more current national sample, supporting a major premise of behavioral economic theory for alcohol use. Additionally, this approach would facilitate comparisons of activity engagement among those who engaged in alcohol use and those who did not during mid-adolescence, which may increase our understanding of early differences in patterns of activity engagement.

The purpose of the current study was to examine differences in engagement with types of activities that may be incompatible with alcohol use, compared among adolescents 15–17 years old who endorsed alcohol use and those who did not. The current study used a nationally representative sample of adolescents from the most recent version (2019) of the Monitoring

the Future study (MTF study) [21]. The current study examined involvement in types of activities (e.g., school-based activities, volunteering with peers) that may be incompatible with and may limit the opportunity for alcohol use, as well as activities that may facilitate the opportunity for alcohol use among adolescents. We selected activities from pre-existing survey items from the MTF study and identified them as incompatible or facilitating based on previous patterns in the literature. In addition to examining differences across those who endorsed alcohol use and those who did not, we also examined the role of activity engagement in frequency of alcohol use among adolescents who endorsed alcohol use.

## Methods

### Study data

Monitoring the Future (MTF) data from 2019 were used in the current study. Each year, the MTF study selects high schools across the United States using a multi-stage process to ensure a nationally representative sample of adolescents in 8[th] grade (age 12–14) and 10[th] grade (age 15–17) [21]. All participants completed survey questions aimed to examine social, behavioral, and environmental influences related to substance use behaviors. Four versions of the surveys were developed for MTF, with each containing core questions related to substance use behaviors. The survey versions differed on their coverage of questions related to social and environmental factors. Participants were randomly assigned to complete one of the four survey versions. Due to the availability of items representing activities that may be incompatible with alcohol use, the current study used data from participants who completed Form 2. Informed consent was provided by parents or guardians of all participants as part of the parent MTF study [21]. We assessed whether measures used in the study performed similarly across the 8[th] and 10[th] grade groups, which could facilitate a direct comparison of activity engagement and alcohol use among younger (8[th] grade) and older (10[th] grade) adolescents. However, our assessment of measurement invariance between the 8[th] and 10[th] grade samples indicated the selected items performed differently between the two groups and therefore could not be directly compared. Thus, 10[th] grade students (age 15–17) were selected as the focus of the study given higher drinking prevalence among 10[th] grade students compared to 8[th] grade students in the sample [22].

### Participants

Form 2 data collection for MTF was conducted with N = 4626 10[th] grade participants. Within this sample, 45% ($N$ = 2096) of participants endorsed alcohol use at least once in their lifetime, whereas 55% ($N$ = 2530) of participants reported no history of alcohol use in their lifetime. Based on lifetime history of alcohol use, participants were categorized as being in the alcohol use or no alcohol use group for analyses.

### Measures

Survey topics from Form 2 contained core questions related to substance use behaviors as well as engagement in a range of activities. To identify items to include in analyses, all items from all pre-existing survey measures were reviewed, and items were included in the study analysis if they measured: 1) engagement in an activity that may be incompatible with alcohol use in the short term (e.g., exercise; volunteering) or longer term (e.g., enjoyment of and engagement in activities at school); and 2) engagement in an activity that may facilitate alcohol use (e.g. spending time with friends unsupervised). Items were classified as potentially incompatible or facilitating based on findings reported in the prior literature. Some activities were included as

exploratory activities and are described in further detail below. For analyses with the alcohol use group, items characterizing direct alcohol use behavior were also included. The identified items were then organized into the following constructs of interest: engagement in types of activities that may be incompatible with alcohol use, activities that may facilitate alcohol use, and alcohol-involved activities (alcohol use group only). Each construct is detailed below, along with internal consistency estimates for Cronbach's alpha and McDonald's omega. Acceptable values were considered to range from .70 to .90 [23,24]. A full list of items included in the current study can be found in S1 Table in S1 File, and a summary of reliability estimates for included items can be found in S2 Table in S1 File.

**Activities that may be incompatible with alcohol use.** Activities that may limit engagement with alcohol use were conceptualized as activities that may be incompatible with alcohol use in the short term (e.g., exercising, volunteering in the community), or the longer term (e.g., school-based activities). Items available in the pre-existing survey measures that characterized types of potentially incompatible activities were as follows: exercise, volunteering, enjoyment of school, employment, going to concerts, and going to the mall. The exercise construct contained two items which captured the frequency of exercise, and had acceptable reliability ($\alpha$ = .80, $\omega$ = .81) for the full sample. The enjoyment from school construct contained four items aimed to capture pleasure derived from and engagement in school. These four items demonstrated acceptable reliability ($\alpha$ = .70, $\omega$ = .73) for the full sample. The remaining activities, consisted of volunteering, employment, going to concerts, and going to the mall, and were all included as single item constructs because there were no other available survey items in the MTF data that aligned with the activities.

**Additional activities.** Additional items were selected from the pre-existing survey measures as exploratory activity items. The constructs were considered to be relevant to the experiences of 10th grade students; however, it was unclear based on theory and the previous literature whether these activities may limit or facilitate engagement in alcohol use. The following constructs were examined: media use, dating, and time spent alone. To measure media use, three items were selected to evaluate the frequency of texting, social media use, and video gaming. The reliability estimates suggested acceptable reliability ($\alpha$ = .66, $\omega$ = .70) for the full sample. Items focused on dating and time spent alone (without parents and friends) were assessed as single items.

**Activities that may facilitate alcohol use.** To measure activities that may facilitate alcohol use, six items were chosen that identified situations that may present the opportunity for alcohol use. However, in contrast to the alcohol-involved items (detailed in section below, relevant for alcohol use group only), alcohol use was not directly addressed in the questions. Items such as "How often do you do each of the following? Go to parties or other social affairs" were selected. The reliability estimates for the construct suggested acceptable reliability ($\alpha$ = .69, $\omega$ = .71) for the full sample. In addition to the complementary items that were socially oriented, a construct of boredom was specified by two items that reflected a lack of available reinforcement from an individual's environment and activities (which may present an elevated risk for alcohol use as an activity). Items consisted of "I am often bored" and "I often find myself with nothing to do." The reliability estimates for this construct suggested good reliability ($\alpha$ = .87, $\omega$ = .87) for the full sample.

**Alcohol-involved activities for alcohol use subsample.** To measure alcohol-involved activities, eight items were selected from surveys that assessed activities directly related to alcohol use. Items such as, "During the last 12 months, how often (if ever) have you used alcohol in each of the following places? At friends' house" were selected. The selected items were only administered to those who endorsed previous alcohol use. The alpha reliability estimate for alcohol-involved activities was 0.83 and the omega reliability was 0.83, suggesting good reliability among the alcohol use group.

**Alcohol use frequency for alcohol use subsample.** Items were selected to measure alcohol use frequency at four different time points: 2 weeks, 30-days, 12-months, and lifetime use. More specifically, the questions inquired about frequency estimates of 30-day, 12-month, and lifetime alcohol use (e.g., "On how many occasions (if any) have you had alcoholic beverages to drink—more than just a few sips"), and frequency of binge drinking episodes within the last two weeks (e.g., "How many times have you had five or more drinks in a row?"). The selected items related to alcohol use frequency were only administered to those who endorsed a history of alcohol use. Reliability estimates indicated good reliability among the alcohol use frequency construct ($\alpha$ = .85, $\omega$ = .85) among the alcohol use group. Items regarding alcohol consequences or related problems were not assessed in Form 2 and thus the construct characterized frequency of alcohol use behavior.

## Statistical analyses

Pearson correlations between the indicator variables included in the model were examined to gain a preliminary understanding of associations among constructs for each group. Then, a measurement model with all constructs (except alcohol constructs) were evaluated using confirmatory factor analysis (CFA) for the full sample [25]. Specifically, 11 activity factors were included in the model: exercise, volunteering, enjoyment of school, employment, going to concerts, going to the mall, media use, dating, time spent alone, facilitating activities, and boredom. The boredom factor as well as the exercise factor were defined by two indicators; thus, tau equivalence was evaluated and met. Tau equivalence assumes that the two indicators have approximately equal reliability and can be assessed by comparing models with and without this constraint enforced [26]. The results of the chi-square difference test comparing these models supported the equality constraint for the boredom construct, but not the exercise construct (see findings below). Thus, tau equivalence was only assumed for the boredom construct. Researchers recommend evaluating this assumption with constructs measured by only two-indicators [25]. To assess model fit, we used the Comparative Fit Index (CFI), Tucker-Lewis Index (TLI), Root Mean Square Error of Approximation (RMSEA) index, and the Standardized Root Mean Square Residual (SRMR) index [25]. A CFI and TLI score greater than .90 is considered acceptable model fit, whereas scores below .90 are considered poor fit [25]. RMSEA and SRMR scores of .05-.08 are considered acceptable, whereas scores ranging from .08-.10 indicate mediocre fit and scores greater than .10 indicate poor fit.

Measurement invariance was assessed between the alcohol use and no alcohol use groups to determine if the activity constructs performed similarly between the two groups before comparing latent means for activity engagement [27]. In order to compare differences in activity engagement between the alcohol use and no alcohol use group, the measurement model is required to pass strong invariance [27]. When comparing the weak invariance to strong invariance model, if the change in CFI did not decrease more than .01 [28] and if the RMSEA indices fell within the confidence intervals [27], the measurement model was deemed to pass strong invariance. Measurement invariance was demonstrated, and structural invariance was assessed to examine differences in latent means and variances between the alcohol use and no alcohol use groups across the activity constructs.

In addition to the analysis by group, we sought to examine the role of activities in alcohol use frequency among adolescents who endorsed alcohol use. Thus, we evaluated a second measurement model, which included the aforementioned activity constructs as well as two additional constructs related to alcohol-involved activities and alcohol use frequency. Because the alcohol-involved activities and alcohol use frequency items were presented to only participants who endorsed a history of alcohol use, this second measurement model was evaluated using

only participants assigned to the alcohol use group. Given that alcohol use behavior at the individual level is strongly correlated over time, one correlated residual was allowed on the alcohol use factor for alcohol use frequency at two weeks and 30 days. Further, two correlated residuals were allowed on the alcohol-involved activities given the similarity in location of alcohol use (e.g., at a friend's house vs. at a party; at a school during the day vs. near school). Next, within a structural equation model (SEM) framework, latent regressions (at an alpha level of .01) were tested to determine if types of incompatible activities (exercise, volunteering, enjoyment of school, employment, going to concerts, and going to the mall), additional activities (media use, dating, and time spent alone), facilitating activities, boredom, and alcohol-involved activities were significantly associated with alcohol use frequency.

Analyses were conducted in R Version 4.0.3 [29], using the R package Lavaan (version 0.6–7) [30] for SEM analyses. To evaluate potential normality violations from skewed responses, the model was evaluated using Maximum Likelihood (ML) and Robust Maximum Likelihood (MLR). Results were consistent across both estimators providing evidence that any minor violations did not influence the model results. Thus, results provided below only include ML estimation [25]. During model specification, the fixed-factor method of scale setting was used to fix latent variances to 1.0. This method of scale setting provides estimations in a standardized metric. In addition, the fixed-factor method provides information regarding between-construct and variance correlations [27]. To determine the extent of missing data, percent missing analyses were conducted in R on each item included in the model. Per the MTF codebook, missing data occured if participants were absent from school during data collection, or transferred to another school or dropped out of school before data collection [21]. Overall, missingness per indicator ranged from 0% to 23%, with a mean percentage of missing data across all variables of 4.2%. To address the missingness, we implemented full information maximum likelihood (FIML) in R. FIML uses observed responses to supplement the loss of information to provide more accurate conclusions regarding the hypothesized models [31]. Using FIML allowed us to include all available participant data in the analyses.

## Results

### Participant characteristics

Demographic characteristics of the sample are presented in Table 1. Approximately half of the participants were female (50%; $n$ = 2301), and 52% of the sample were at least 16 years of age. The sample consisted primarily of participants who were White (48%), Black (12%), or Hispanic (19%), and approximately 21% of the participants did not provide their racial or ethnic identity. When examining the sample composition across the alcohol use and no alcohol use groups, results from a chi-square difference test revealed that the alcohol use group had significantly fewer male participants, $\chi^2$ (1) = 10.44, $p < .001$ compared to the no alcohol group. Further, findings from a one-way analysis of variance revealed that the alcohol group had fewer participants identifying as White, $F(1) = 13.45$, p < .001, compared to the no alcohol use group. The two groups did not significantly differ in age.

### Correlation matrix

We examined Pearson correlation matrices of the indicators used in the analyses for the no alcohol use group (S3 Table in S1 File) and the alcohol use group (S4 Table in S1 File). In the no alcohol use group, incompatible activities such as enjoyment from school were negatively correlated with facilitating activities such as unsupervised time with peers on school nights and boredom. In the alcohol group, incompatible activities such as volunteering were negatively correlated with alcohol use frequency items. Facilitating activities were positively

**Table 1. Demographic characteristics of full sample and alcohol vs. non-alcohol groups.**

| | | Full Sample (N = 4626) | | Alcohol Use Group (N = 2096) | | No Alcohol Group (N = 2530) | |
|---|---|---|---|---|---|---|---|
| **Variable** | | **N** | **%** | **N** | **%** | **N** | **%** |
| Sex | | | | | | | |
| | Female | 2301 | 49.7% | 1097 | 52.3% | 1204 | 50.3% |
| | Male | 2231 | 48.2% | 957 | 45.7% | 1274 | 47.6% |
| | Missing | 94 | 2.1% | 42 | 2.0% | 52 | 2.1% |
| Race/Ethnicity* | | | | | | | |
| | White | 2224 | 48.1% | 1075 | 51.3% | 1149 | 45.4% |
| | Black | 546 | 11.8% | 179 | 8.5% | 367 | 14.5% |
| | Hispanic | 871 | 18.8% | 394 | 18.8% | 477 | 18.9% |
| | Missing | 985 | 21.3% | 448 | 21.4% | 537 | 21.2% |
| Age** | | | | | | | |
| | Under 16 years | 2136 | 46.1% | 948 | 45.2% | 1188 | 47.0% |
| | 16 years or older | 2425 | 52.4% | 1120 | 53.4% | 1305 | 51.6% |
| | Missing | 65 | 1.5% | 28 | 1.4% | 37 | 1.4% |

*MTF survey did not provide data for other race/ethnicity identities.

**MTF survey provided age as below 16 years, or 16 years and older.

correlated with alcohol use frequency; however, boredom was negatively correlated with alcohol use frequency.

## Confirmatory factor analysis for the measurement model: Full sample

For the full sample, the hypothesized measurement model was specified using available survey items to characterize the following incompatible activities as latent constructs: exercise, volunteering, enjoyment of school, employment, going to concerts, and going to the mall. Additional activities were also examined as latent constructs, which included: media use, dating, and time spent alone. Facilitating activities and boredom were modeled as independent latent constructs. A summary of the findings can be found in S5 Table in S1 File. Overall, the standardized factor loadings across the constructs ranged from .80 to .84 for exercise, .32 to .82 for enjoyment from school, .33 to .83 for media use, .36 to .67 for facilitating activities, and .86 to .90 for boredom. Because the boredom construct and the exercise construct consisted of two indicators each, tau equivalence was assumed, and the factor loadings were equated. To test if the assumption of tau equivalence was tenable, models with and without these constraints were evaluated. A chi-square difference test revealed that the model passed the tau equivalence assumption for the boredom construct ($\Delta\chi^2 = 1.15$, $p = .28$), and did not pass the tau equivalence assumption for the exercise construct ($\Delta\chi^2 = 41.43$, $p < .001$). Tau equivalence was assumed for only the boredom construct. Fit statistics for the measurement model indicated acceptable model fit [25] with RMSEA = .043 (90% CI = .041-.045), SRMR = .035, TLI = .905, CFI = .932, and $\chi^2$ (182) = 1723.44, $p < .001$.

**Table 2. Model fit indices for the measurement invariance models (N = 4626).**

| Model | $\chi^2$ | df | p | CFI | TLI | RMSEA | 90% CI | SRMR | Pass? |
|---|---|---|---|---|---|---|---|---|---|
| Configural | 1455.23 | 365 | < .001 | .929 | .902 | .045 | .042-.047 | .038 | Yes |
| Weak | 1474.03 | 375 | < .001 | .929 | .904 | .044 | .042-.047 | .039 | Yes |
| Strong | 1592.18 | 386 | < .001 | .928 | .898 | .046 | .043-.048 | .040 | Yes |

## Measurement invariance: Alcohol use vs. no alcohol use group

To determine whether the activity items measured the same underlying construct for both the alcohol use and no alcohol use groups, we evaluated measurement invariance. Table 2 provides the fit indices for the nested sequence in the multiple-group CFA. The configural invariance model demonstrated acceptable model fit (see Table 2). After adding constraints on the factor loadings to evaluate weak invariance, the change in CFI from the configural to weak model was less than .001, which provides evidence of weak factorial invariance [28]. The strong invariance model demonstrated acceptable model fit as well (RMSEA = .046; CFI = .928; TLI = .898; SRMR = .040; $\chi^2$ (386) = 1592.18, $p <$ .001). Further, the change in CFI from the weak invariance to strong invariance models remained less than .001, and thus, the model met current recommended guidelines for strong invariance. As a result, the activity items showed a similar factor structure for the alcohol and no alcohol use groups, and latent mean differences in activity engagement could be evaluated.

## Latent mean differences in activity engagement: Alcohol use vs. no alcohol use group

To evaluate latent mean differences across both the alcohol use and no alcohol use group in engagement with activities, equality constraints were placed on the latent means and the constrained model was compared to the strong invariance model. Findings revealed that there were significant differences in activity engagement between the two groups ($\Delta\chi^2 = 316.90$, $p <$ .001). Sequential pairwise comparisons revealed that participants in the no alcohol use group reported higher engagement in activities that may be incompatible with alcohol use, specifically enjoyment from school ($p = .001$) and going to the mall ($p < .001$). Participants in the alcohol use group reported higher levels of engagement in facilitating activities ($p < .001$), dating ($p < .001$), boredom ($p = .001$), and media use ($p < .001$). Observed latent effect sizes were small in magnitude for all significant findings. There were no significant differences in volunteering, exercise, employment, or leisure time alone between the two groups. A summary of these findings is presented in Table 3.

**Table 3. Comparison of latent mean differences between alcohol use and no alcohol use groups.**

| Latent Variable | Latent Mean Difference | $p$ | Subsample with higher engagement | Latent Cohen's D |
|---|---|---|---|---|
| Incompatible Activities: | | | | |
| Exercise | -.001 | .976 | - | .01 |
| Volunteer | -.059 | .096 | - | .06 |
| Enjoyment from School | -.136 | .001 | No Alcohol | .14 |
| Employment | .042 | .224 | - | .04 |
| Concerts | .105 | .002 | Alcohol | .10 |
| Mall | -.208 | < .001 | No Alcohol | .22 |
| Additional Activities: | | | | |
| Media use | .183 | < .001 | Alcohol | .18 |
| Dating | .163 | < .001 | Alcohol | .15 |
| Time spent alone | .042 | .260 | - | .04 |
| Facilitating Activities: | | | | |
| Facilitating | .392 | < .001 | Alcohol | .39 |
| Boredom | .116 | < .001 | Alcohol | .12 |

(-) = No significant differences in latent means between subsamples.

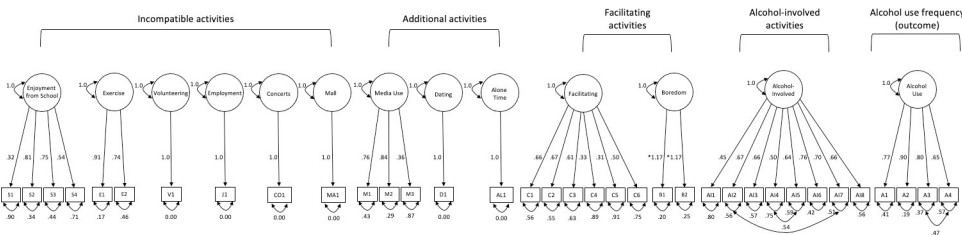

**Fig 1. Hypothesized measurement model for alcohol use subsample (n = 2096).** This figure represents the hypothesized constructs for patterns of activity engagement among adolescents. Each construct represents activities which may be incompatible with alcohol use or facilitate alcohol use. Latent covariances are not included in the figure.

## Confirmatory factor analysis for the measurement model: Alcohol use subsample

The second hypothesized measurement model was specified using the latent constructs from the previous model to characterize the following incompatible activities as latent constructs: exercise, volunteering, enjoyment of school, employment, going to concerts, and going to the mall. Additional activities were also examined as latent constructs, which included: media use, dating, and time spent alone. Facilitating activities and boredom were modeled as independent latent constructs as well. Additionally, available survey items which directly mentioned activities related to alcohol use as well as alcohol use frequency were also examined. A summary of the findings can be found in Fig 1 below and in S6 Table in S1 File. Overall, the standardized factor loadings across the constructs ranged from .73 to .91 for exercise, .32 to .81 for enjoyment from school, .36 to .86 for media use, .30 to .67 for facilitating activities, .87 to .90 for boredom, .45 to .76 for alcohol-involved activities, and .65 to .90 for alcohol use frequency. Because the boredom construct and the exercise construct consisted of two indicators each, tau equivalence was assumed, and the factor loadings were equated. To test if the assumption of tau equivalence was tenable, models with and without these constraints were evaluated. A chi-square difference test revealed that the model passed the tau equivalence assumption for the boredom construct ($\Delta\chi^2$ = 2.07, $p$ = .15) and did not pass the tau equivalence assumption for the exercise construct ($\Delta\chi^2$ = 28.41, $p < .001$). Thus, tau equivalence was only assumed for the boredom construct. Fit statistics for the measurement model indicated acceptable model fit [25] with RMSEA = .044 (90% CI = .043-.046), SRMR = .044, TLI = .893, CFI = .913, and $\chi^2$ (486) = 2456.35, $p < .001$.

## Structural equation model

To determine if the incompatible activities (exercise, volunteering, enjoyment of school, employment, going to concerts, and going to the mall), additional activities (media use, dating, and time spent alone), facilitating activities, and alcohol-involved activities predicted alcohol use frequency, a latent regression model was tested using SEM. Fit statistics for the structural equation model indicated acceptable model fit [25] with RMSEA = .044 (90% CI = .043-.046), SRMR = .044, TLI = .893, CFI = .913, and $\chi^2$ (486) = 2456.35, $p < .001$. At an alpha level of .01, results indicated that going to the mall ($\beta$ = -.07, $p$ = .002) was negatively associated with alcohol use frequency. Enjoyment from school ($\beta$ = -.06, $p$ = .012) was negatively associated with alcohol use frequency at alpha level of .05. Other potential incompatible activities such as exercise, volunteering, employment, and going to concerts were not significantly associated with alcohol use frequency (Table 4). While boredom was expected to facilitate alcohol use, results revealed that boredom was negatively associated with alcohol use frequency ($\beta$ = -.06, $p$ = .01),

**Table 4. Regression estimates predicting alcohol use frequency among alcohol use group.**

| Predictor | | Std. Beta | Std. Error | *p* | Confidence Interval |
|---|---|---|---|---|---|
| Incompatible Activities: | | | | | |
| | Exercise | -0.010 | 0.038 | .675 | -0.06–0.09 |
| | Volunteer | -0.033 | 0.031 | .086 | -0.11–0.01 |
| | Enjoyment from School | -0.059 | 0.037 | .012 | -0.17 – -0.02 |
| | Employment | 0.030 | 0.030 | .116 | -0.01–0.11 |
| | Concerts | -0.014 | 0.031 | .477 | -0.08–0.04 |
| | Mall | -0.068 | 0.034 | .002 | -0.17 – -0.04 |
| Additional Activities: | | | | | |
| | Media use | -0.020 | 0.036 | .378 | -0.10–0.04 |
| | Dating | -0.043 | 0.033 | .041 | -0.13 – -0.003 |
| | Time spent alone | -0.033 | 0.029 | .076 | -0.11–0.01 |
| Facilitating Activities: | | | | | |
| | Facilitating | 0.180 | 0.056 | < .001 | 0.02–0.39 |
| | Boredom | -0.059 | 0.038 | .011 | -0.17 – -0.02 |
| Alcohol-involved Activities: | | | | | |
| | Alcohol-involved | 0.686 | 0.057 | < .001 | 0.97–1.19 |

indicating that individuals who reported greater boredom were engaged in less frequent alcohol use. Regarding additional activities (media use, dating, and time spent alone), none of the activities were associated with alcohol use frequency at an alpha level of .01. However, dating was negatively associated with alcohol use frequency ($\beta$ = -.04, $p$ = .041) at an alpha level of .05. Finally, results indicated that facilitating activities ($\beta$ = .18, $p <$ .001) and alcohol-involved activities ($\beta$ = .69, $p <$ .001) were positively associated with alcohol use frequency. Results with standardized parameter estimates are summarized in Table 4. Fig 2 presents the complete structural equation model with both significant and non-significant paths.

Non-significant paths were trimmed from the regression model. Fit statistics for the trimmed structural equation model indicated acceptable model fit (Kline, 2016) with RMSEA = .059 (90% CI = .056-.061), SRMR = .052, TLI = .908, CFI = .922, and $\chi^2$ (178) = 1460.82, $p <$ .001. The final model is summarized in Table 5 and presented in Fig 3. Overall, frequency of going to the mall was associated with less frequent alcohol use. Additionally, higher boredom was associated with less frequent alcohol use. Conversely, greater engagement in facilitating activities and alcohol-involved activities was associated with greater alcohol use frequency.

## Discussion

Behavioral economic theory indicates that alcohol use among adolescents may occur when adolescents engage in few activities that serve as alternatives to alcohol use [10–13]. However, the types of activities that may limit engagement with alcohol use in adolescence are largely unclear. The current study applied behavioral economic theory to examine alcohol use patterns among adolescents from a public health lens. The study leveraged the availability of an existing nationally representative data source and tested the differences in engagement in types of activities among adolescents 15–17 years old who endorsed alcohol use relative to adolescents who did not use alcohol. Overall, there were differences in the patterns of activity engagement across adolescents who did and did not use alcohol. Adolescents who did not use alcohol reported significantly higher engagement in some activities that may be incompatible with alcohol use, including school activity engagement and enjoyment. In contrast, adolescents who used alcohol had significantly higher levels of engagement in activities that may facilitate

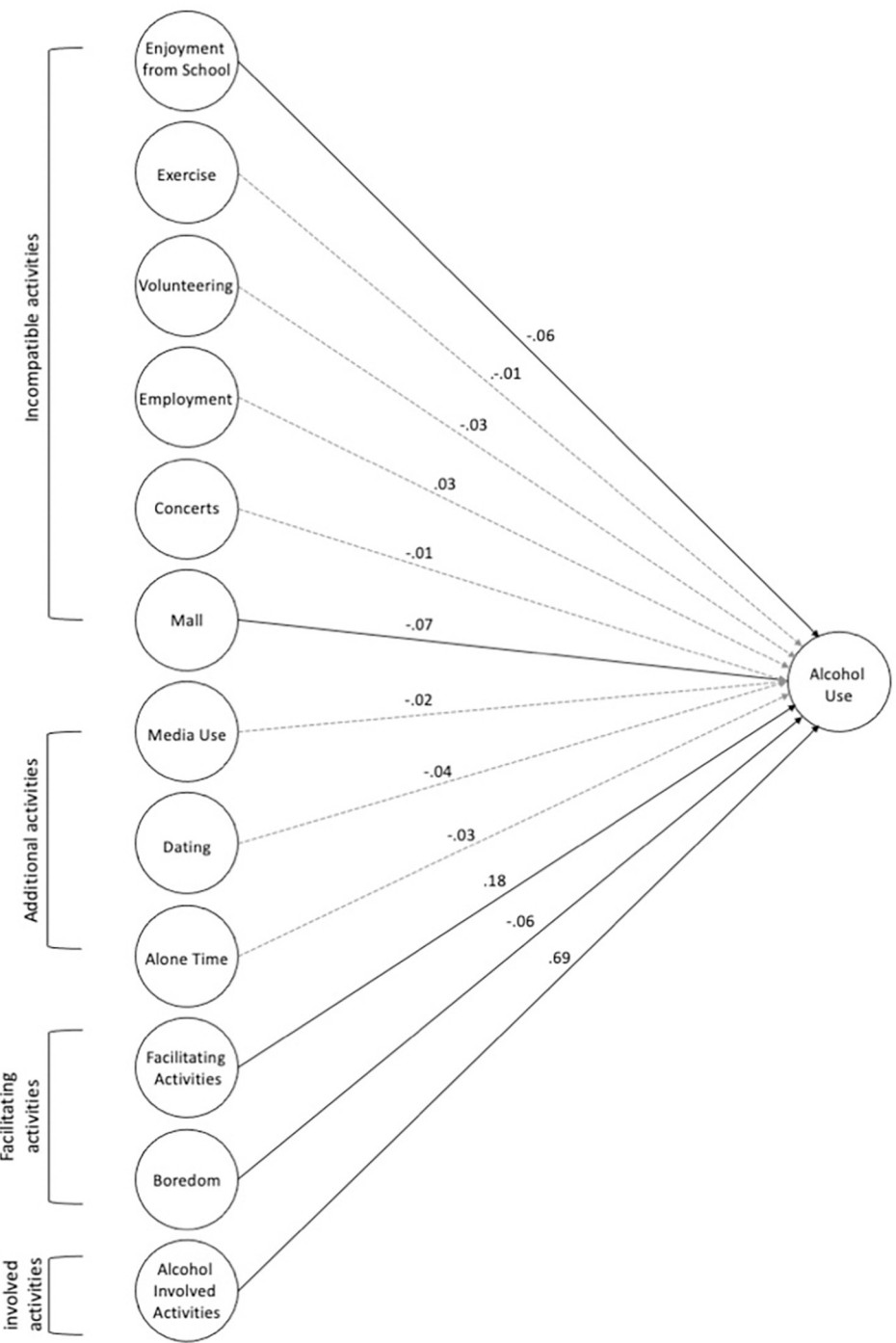

**Fig 2. Full structural equation model for alcohol use subsample (n = 2096).** This figure presents the significant and non-significant paths predicting alcohol use frequency. Significant paths are represented as solid lines using p = .01 as the cut-off. Non-significant paths are presented as dashed lines. Provided values are the standardized betas. Confidence intervals and p-values are summarized in Table 4. Indicators and latent covariances are not included in figure.

alcohol use (e.g., unsupervised time with friends, etc.), as well as activities that may yield more exposure to alcohol (e.g., dating, media use). In addition, when examined among the subsample of adolescents who used alcohol, school engagement was negatively associated with

**Table 5. Trimmed regression estimates predicting alcohol use frequency.**

| Predictor | Std. Beta | Std. Error | p | Confidence Interval |
|---|---|---|---|---|
| Mall | -0.077 | 0.033 | < .001 | -0.19 – -0.05 |
| Boredom | -0.051 | 0.034 | .023 | -0.15 – -0.01 |
| Facilitating | 0.150 | 0.047 | < .001 | 0.14–0.32 |
| Alcohol-involved | 1.078 | 0.055 | < .001 | 0.97–1.19 |

frequency of alcohol use, further supporting the role of school involvement as a factor that may be associated with lower alcohol use among adolescents. Overall, the findings partially support the premise of behavioral economics related to activity engagement; findings identified one key factor that may be incompatible with alcohol use frequency and confirmed the premise that engagement in facilitating activities may promote alcohol use frequency among a national sample of adolescents.

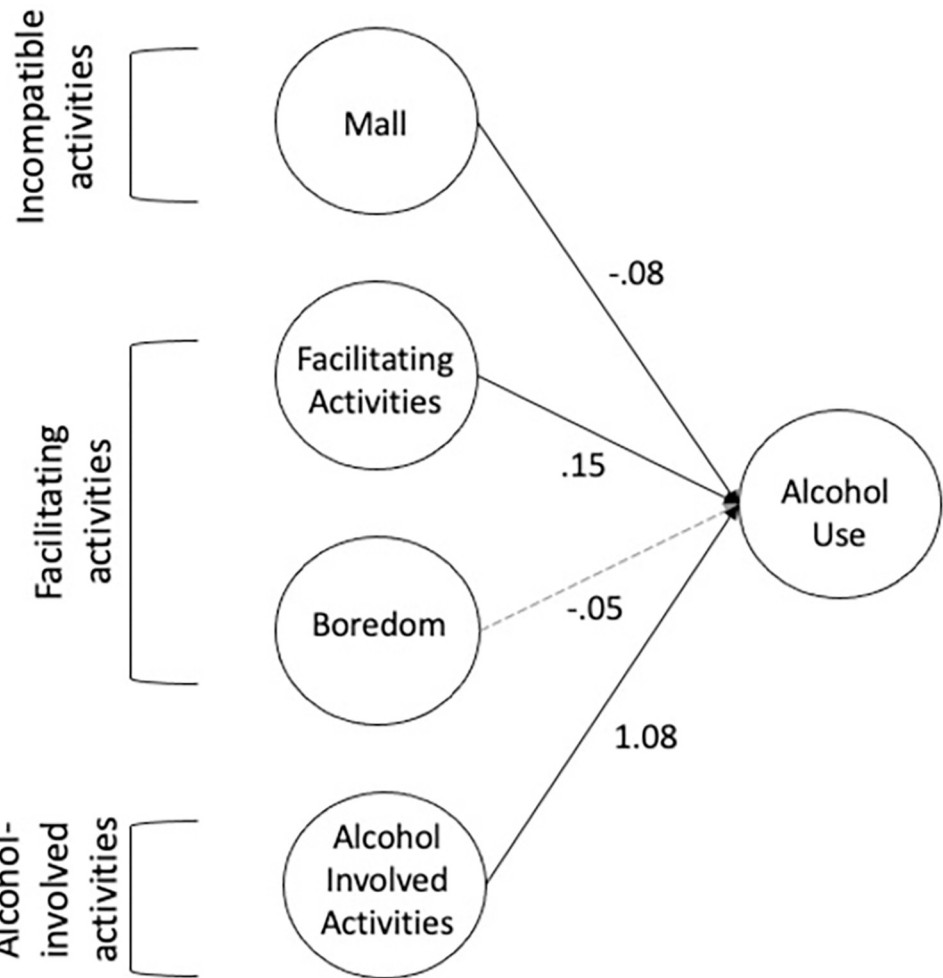

**Fig 3. Trimmed structural equation model for alcohol use subsample (n = 2096).** After trimming non-significant paths from structural equation model, four constructs remained which predicted alcohol use frequency. Significant paths at p = .01 are represented using solid lines. Significant paths at p = .05 are represented using dashed lines. Provided values are the standardized betas. Confidence intervals and p-values are summarized in Table 5. Indicators and latent covariances are not included in figure.

Overall, findings provide some support for premise that engagement in alternative activities to alcohol use may be associated with lower alcohol use among adolescents. Adolescents who did not use alcohol endorsed greater enjoyment of and engagement with school activities, as well as going to the mall. Enjoyment and engagement with school activities may be incompatible with alcohol use in the long term, such that adolescents may invest substantial effort and time in school, which may be incompatible with alcohol use both in terms of a time investment (e.g., investing time in studying vs. drinking with friends) and in terms of the effects that alcohol involvement may have on school engagement (e.g., drowsiness in class following a night of drinking, etc.). This premise is also supported by the subgroup analyses with adolescents who endorsed alcohol use, which revealed a negative association between school engagement and frequency of alcohol use. Thus, the findings indicated that engagement with school may be broadly incompatible with alcohol use and was associated with: 1) no lifetime drinking; and 2) (lower) frequency of alcohol use (among those who did drink alcohol). However, despite the findings related to school involvement, the only other incompatible activity identified was going to the mall, such that adolescents who did not use alcohol endorsed going to the mall more frequently, and going the mall was negatively correlated with frequency of alcohol use among those who used alcohol. It may be that malls have a strong adult presence (in stores as workers, shoppers, etc.), and may have limited sales of alcohol, factors that may be reduce the likelihood of alcohol use. However, given the large decline in popularity of malls in the US in recent decades [32], the role of malls in current adolescents' lives is somewhat unclear. It may be that adolescents who go to the mall have other characteristics that may also limit alcohol use. Finally, many other potentially incompatible activities were examined, including volunteering in the community, exercising, and working for pay; however, no other activities were associated with alcohol involvement. Overall, engagement in school and going to the mall were identified as potentially incompatible activities associated with limited alcohol involvement. More work is needed to further delineate activities that may be associated with lower alcohol use among national samples of adolescents to inform prevention efforts.

Findings overall lend support to the behavioral economics premise that engagement with activities that may facilitate alcohol use may yield greater alcohol use. Consistent with the existing literature that examined regional samples of adolescents, the present study provides evidence that greater engagement with facilitating activities may be associated with more frequent alcohol use when examined using a large, national sample of US adolescents. In addition to facilitating activities, adolescents also engaged in activities that may increase exposure to alcohol (e.g., dating, parties with peers), and alcohol-endorsing media content (on social media, etc.). Thus, adolescents who endorsed alcohol use engaged in a pattern of activities that were distinct and likely serve to increase the likelihood of alcohol involvement. In addition, previous research has demonstrated that higher boredom among adolescents was positively associated with alcohol use [33–36]. However, when examining the association between boredom and alcohol use frequency among the alcohol use group, findings revealed a negative correlation between boredom and alcohol use, inconsistent with the behavioral economics assumption. It may be that adolescents who use alcohol and who reported higher boredom may have restricted access to socially-based activities that may facilitate alcohol use, resulting in a lack of engagement with environmental activities broadly. However, more work may be required to understand the role of boredom in adolescent alcohol use.

The current study adds to the existing literature by examining patterns of activity engagement and alcohol use during mid-adolescence (~15–17 years). Mid-adolescence is a developmental period in which adolescents gain greater independence in choosing how they spend their time, which could be allocated to school-related activities, extracurricular activities (sports, clubs, etc.), and/or to alcohol use and other risky behaviors. Furthermore, in the

United States, mid-adolescence marks the age in which many adolescents obtain a driver's license, which serves to increase their independence and ability to transport themselves without parental supervision. Additionally, during mid-adolescence, greater emphasis is placed on peer approval [37,38]. Accordingly, there is substantial evidence to suggest adolescents are more likely to consume alcohol if their friends also drink [38–43], and most drinking behavior occurs in the presence of friends at parties [44]. Thus, this juncture of increased mobility, independence, and peer influences during mid-adolescence may therefore contribute to greater engagement in activities with provide greater opportunity for alcohol use. Current findings from the initial regression and trimmed regression models are consistent with previous literature, such that greater engagement in facilitating activities (i.e., socializing with peers, attending parties without supervision) as well as alcohol-involved activities were associated with greater alcohol use among mid-adolescents.

Overall, our study represents an application of a behavioral science premise to understand alcohol use patterns among adolescents from a public health perspective. Despite the potential utility of applying behavioral science principles and hypotheses to understand population-level behavior, few studies have attempted to apply behavioral economic principles to public health-related topics. Thus, our study has attempted to bridge this gap in relation to behavioral economics and adolescent alcohol use. However, no nationally representative study of adolescents, including the MTF data we used in our analyses, has included established behavioral survey measures of engagement in alternative reinforcing activities to alcohol use. Thus, we used a novel approach of identifying potential activities for analysis by collating items from pre-existing survey measures used in the study and analyzing types of activities. However, few activities that were potentially incompatible with alcohol use were identified as being significantly associated with alcohol use. Future waves of the MTF study and other national survey studies may consider including an established survey measure that characterizes alcohol-involved and alcohol-free activities as potential sources of reinforcement. The study did, however, provide an examination of facilitating activities and findings overall provided support for the premise of behavioral economics and the prior literature that engagement in complementary activities may be associated with greater alcohol use frequency.

The current study had limitations. First, given the cross-sectional design of the current study, no conclusions may be made regarding causality of activity engagement and adolescent alcohol use. Future longitudinal studies are needed to assess whether engagement in incompatible activities may limit or prevent alcohol use throughout adolescence. In addition, the current findings relied upon self-reported alcohol use, which may be prone to underestimation. However, using self-report items facilitated data collection with a nationally representative sample and likely enhanced the generalizability of the findings. Also, previous research has demonstrated motives for alcohol use may differ among individuals who engage in light drinking vs. heavy drinking [45], which the current study was unable to examine due to available data. Future research may examine if patterns of activity engagement differ across light vs. heavy drinking patterns. Further, the current study was limited due to the available items from the MTF study and may not fully capture all possible activities that may be incompatible with or facilitate alcohol use. Future research may select more types of activities to further identify which may be incompatible with alcohol use. The current study was also unable to determine if involvement in some activities was compulsory (e.g., required for athletics team participation) or voluntary; therefore, future research may determine if required activities may similarly limit alcohol involvement relative to voluntary activities among adolescents. Finally, the current study examined engagement in activities, but did not examine adolescent attitudes towards activities, social norms, and perceived control over their involvement in these activities. Future research may wish to use the Integrated Behavioral Model [46] to better

understand patterns of activity engagement as well as underlying motivations for participating in these activities.

## Conclusion

Our study represents an application of a behavioral economics premise to understand alcohol use patterns among adolescents from a public health lens. Overall, analyses of a national sample of US adolescents revealed substantial differences in patterns of activity engagement across those who did not use alcohol and those who did. Adolescents who did not use alcohol reported higher engagement in some activities that may be incompatible with alcohol use (e.g., school enjoyment and engagement); however, endorsement of most potentially incompatible activities was not significantly higher among adolescents who did not use alcohol, relative to those who did. In contrast, adolescents who used alcohol had significantly higher levels of engagement in activities that may facilitate or yield more exposure to alcohol use (e.g., unsupervised time with friends, dating, etc.). Findings provided some support for a premise of behavioral economic theory and indicated that greater engagement in facilitating activities was associated with higher levels of alcohol use. More work is needed to identify activities that may be incompatible with alcohol use in mid-adolescence, which is needed to inform targets for prevention programs.

## Supporting information

**S1 File. Contains supporting tables.**
(DOCX)

## Author Contributions

**Conceptualization:** Cassandra A. Sutton, Tera L. Fazzino.

**Formal analysis:** Cassandra A. Sutton.

**Methodology:** Cassandra A. Sutton, Elizabeth Grandfield, Richard Yi, Tera L. Fazzino.

**Supervision:** Elizabeth Grandfield, Tera L. Fazzino.

**Writing – original draft:** Cassandra A. Sutton.

**Writing – review & editing:** Elizabeth Grandfield, Richard Yi, Tera L. Fazzino.

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
