## [Decision Letter · Decision Letter 0]

23 May 2023

PONE-D-23-02338Engagement in types of activities and frequency of alcohol use in a national sample of United States adolescentsPLOS ONE

Dear Dr. Fazzino,

Thank you for submitting your manuscript to PLOS ONE. After careful consideration, we feel that it has merit but does not fully meet PLOS ONE’s publication criteria as it currently stands. Therefore, we invite you to submit a revised version of the manuscript that addresses the points raised during the review process.

We look forward to receiving your revised manuscript.

Kind regards,

Chiranjivi Adhikari, MPH, MHEd., PhD Candidate

Academic Editor

PLOS ONE

Journal Requirements:

3. Thank you for including your ethics statement: "All participants in the main Monitoring the Future study provided written informed consent. Deidentified data were used in the current study."  

For studies reporting research involving human participants, PLOS ONE requires authors to confirm that this specific study was reviewed and approved by an institutional review board (ethics committee) before the study began. Please provide the specific name of the ethics committee/IRB that approved your study, or explain why you did not seek approval in this case.

Additional Editor Comments:

Dear Authors,

Thank you for a rigorous scientific piece of work in behavioral economics, a quite less addressed and worked out areas in public health and behavioral sciences, including its application in different fields. The study is well-designed, carried out with appropriate sample size and analysis. The tools used are within reliable ranges, the model fitting statistics are acceptable ranges. Nonetheless, along with observing and revering to the comments from our reviewers, it would be great if you go with the following points:

1. Alcohol, and its addiction, although has been explored and discussed in a social psychological paradigm, praiseworthily, the coping and emotional nexus of the abuse may be deemed necessary for further discussion. This may be further discussed with drinking-to-cope pathways, as illustrated by Dora et.al (Dora J, Kuczynski AM, Schultz ME, Acuff SF, Murphy JG, King KM. An experimental investigation into the effect of negative affect on the behavioral economic demand for alcohol. Psychology of Addictive Behaviors. 2022 Feb 24.), such as alcohol demand after negative mood, and even stronger among the heavy drinkers. So, this may even demand to furhter explore with heavy and lighter drinkers perspective, if possible. Similarly, higher coping motives and negative urgency may also demand the abuse hgher.

2. The trimmed factors (four) may be discussed furhter with the similarities among them, as what social-cognitions, or cognitive-emotional factors interact, or is there similar pathways for not taking alchol or for those factors, facilitating to take the abuse.

3. Are there any developmental factors, which may have influence in alcoholism, and avoiding ? Not found addressed in discussion, except mentioning about mid-adolescence elsewhere.

4. Both figures (1 & 2) may be needed to further explained aided with brief interpretation of the statistics.

5. Finally, an additional figure is expected, which may illustrate the whole idea, and the variables found significant, and also non-significant, if important, from the literatures, specific to the study.

With regards,

AE

Reviewers' comments:

Reviewer's Responses to Questions

**Comments to the Author**

1. Is the manuscript technically sound, and do the data support the conclusions?

Reviewer #1: Yes

Reviewer #2: Yes

Reviewer #3: Yes

Reviewer #4: Yes

Reviewer #5: Yes

2. Has the statistical analysis been performed appropriately and rigorously? 

Reviewer #1: Yes

Reviewer #2: Yes

Reviewer #3: Yes

Reviewer #4: Yes

Reviewer #5: I Don't Know

3. Have the authors made all data underlying the findings in their manuscript fully available?

Reviewer #1: Yes

Reviewer #2: Yes

Reviewer #3: Yes

Reviewer #4: Yes

Reviewer #5: Yes

4. Is the manuscript presented in an intelligible fashion and written in standard English?

Reviewer #1: Yes

Reviewer #2: Yes

Reviewer #3: Yes

Reviewer #4: Yes

Reviewer #5: Yes

5. Review Comments to the Author

Reviewer #1: Reviewer has no such recommendation for this unique paper. This work is technically sound and covers all important measures. But few things like figure required modification and it should be more informative to all readers.

Reviewer #2: The article was very well written. Overall a good attempt. As the data was carried out from cross-sectional surveys - it majorly suggest of substantial differences in adolescent patterns of engagement. And you carries out really well. however, I wish you to carry out a longitudinal study to generate better evidence for it.

The data covers overall adolescent group. It would be better if you've seperated it to few sub-groups like early adolescent, mid-adolescent and late. (10-13, 14-16, 17-19). But a good to go work..

really appreciate the efforts you put in.

Reviewer #3: Dear Author,

Your work is appreciable, I went through the manuscript with minor corrections to be incorporated in the article. Saying so, the selection of topic is of importance, alcohol consumption as a minor may lead to violence in the tender age, long term physical and mental demormity, and may be a cause of gun violence in USA!

Few, recommendation I would like to put forwards:

1. In page number 22, row number 456, table 5. Trimmed regression estimated predicting alcohol use frequency, the predictor Alcohol-involved is showing confidence intervel of 0.97-1.19, whereas the p value is significant. Could you explain this?

2. In the discussion section few sentences are very lengthy (row number 477-480), please make such sentences small for easy understanding by the readers.

3. I felt that limited references were used in the discussion sections (32 to 36) to strenghten your results. I would sugest to incorporate few more studies to improve the vigour of your paper. Please use the the following study (if apt)!

a) Love My Body: Pilot Study to Understand Reproductive Health Vulnerabilities in Adolescent Girls, doi:10.2196/16336

4. Please format all the references in simillar way, as reference 1 to 9 is arranged differently than the rest.

Please further strenghten your conclusion, as the study is provifding some important outcomes, but the same is not getting reflected in your discussoin and recommendations.

All the best!

Reviewer #4: The manuscript is technically sound and all the data support the conclusions. All the statistical analysis been performed appropriately and rigorously. All the data is verified and free of plagiarism and written in an intelligible fashion and written in standard English.

Reviewer #5: well written peice of Manuscript

I ahev few queires and suggestion for betterment of your manuscript.

1. why you just take additional data such as media use, dating, and time spent alone only? There are even more variables which may imcompactible of alcohol use.

2. In result section you have mentioned data if missing variable. how do you define it ?

3. Along wuth behavioural economic thoey, this manuscript result minghr also supported by intergrated behavioural model. (Please refer it to.. )

5. If you make cut off and alalysis om three categories of adolscent the result might look interstening; early adolscence, middle and late adolscence

6. PLOS authors have the option to publish the peer review history of their article (what does this mean?). If published, this will include your full peer review and any attached files.

Reviewer #1: **Yes: **Biplab Bikash Panigrahi

Reviewer #2: No

Reviewer #3: **Yes: **Shubham Sharma

Reviewer #4: **Yes: **Aakansha Shukla

Reviewer #5: No

While revising your submission, please upload your figure files to the Preflight Analysis and Conversion Engine (PACE) digital diagnostic tool, https://pacev2.apexcovantage.com/. PACE helps ensure that figures meet PLOS requirements. To use PACE, you must first register as a user. Registration is free. Then, login and navigate to the UPLOAD tab, where you will find detailed instructions on how to use the tool. If you encounter any issues or have any questions when using PACE, please email PLOS at figures@plos.org. Please note that Supporting Information files do not need this step.<quillbot-extension-portal></quillbot-extension-portal>

---

## [Author Response · Author response to Decision Letter 0]

16 Aug 2023

Thank you for the opportunity to revise and resubmit this manuscript to PLOS ONE. Revisions were made based on reviewer feedback and we believe this has resulted in a stronger manuscript. An item by item response is presented below. 

Comments from the Editors:

1. Alcohol, and its addiction, although has been explored and discussed in a social psychological paradigm, praiseworthily, the coping and emotional nexus of the abuse may be deemed necessary for further discussion. This may be further discussed with drinking-to-cope pathways, as illustrated by Dora et.al (Dora J, Kuczynski AM, Schultz ME, Acuff SF, Murphy JG, King KM. An experimental investigation into the effect of negative affect on the behavioral economic demand for alcohol. Psychology of Addictive Behaviors. 2022 Feb 24.), such as alcohol demand after negative mood, and even stronger among the heavy drinkers. So, this may even demand to further explore with heavy and lighter drinkers perspective, if possible. Similarly, higher coping motives and negative urgency may also demand the abuse higher.

Response: Due to limitations in the types of drinking data that were available in the national survey, we were unable to explore differences in heavy versus light drinking patterns. The provided survey items only captured the presence of drinking behavior and drinking frequency, and did not capture variability in quantity. We have added this as a possible future direction to the discussion section.

2. The trimmed factors (four) may be discussed further with the similarities among them, as what social-cognitions, or cognitive-emotional factors interact, or is there similar pathways for not taking alcohol or for those factors, facilitating to take the abuse.

Response: A discussion of the trimmed factors model was added to the discussion section. 

3. Are there any developmental factors, which may have influence in alcoholism, and avoiding ? Not found addressed in discussion, except mentioning about mid-adolescence elsewhere.

Response: We have added a discussion about developmental factors related to findings from the current study to the discussion section.

4. Both figures (1 & 2) may be needed to further explained aided with brief interpretation of the statistics.

Response: Captions have been added to both figures to provide further description to readers.

5. Finally, an additional figure is expected, which may illustrate the whole idea, and the variables found significant, and also non-significant, if important, from the literatures, specific to the study.

Response: A figure has been created to illustrate the expected predictors of alcohol use alongside the significant findings (see Figure 2).

Reviewer #1

Reviewer has no such recommendation for this unique paper. This work is technically sound and covers all important measures. But few things like figure required modification and it should be more informative to all readers.

Response: Captions have been added to both figures to provide further description to readers.

Reviewer #2

however, I wish you to carry out a longitudinal study to generate better evidence for it.

Response: We agree that a longitudinal study would be beneficial to capture whether patterns of engagement are predictive of future alcohol use among adolescents. However, we are not able to report on longitudinal findings given the cross-sectional structure of the available data. We have added a suggestion for future research to use longitudinal designs to our discussion section. 

The data covers overall adolescent group. It would be better if you've seperated it to few sub-groups like early adolescent, mid-adolescent and late. (10-13, 14-16, 17-19). 

Response: We appreciate this suggestion; however, unfortunately we were unable to examine adolescents by sub-groups due to limitations in the available data from the national parent survey. The available data only included participants who were 12-14 years old (enrolled in 8th grade) and 15-17 years old (enrolled in 10th grade). We did assess for measurement invariance between the 8th and 10th grade samples to determine if we could compare the samples directly. However, findings indicated the survey questions performed differently between the two groups, and thus we could not make direct comparisons between age groups for reasons outlined in Little (2013) and Schmitt & Kuljanin (2008). Thus, we selected the 10th grade sample (~15-17 years old) as the focus of the study, given the higher drinking prevalence among the 15-17 years old/10th grade group. We had added information regarding this decision to the methods section. 

References:

Little TD. Longitudinal structural equation modeling. New York: The Guilford Press; 2013. 386 p. (Methodology in the social sciences).

Schmitt N, Kuljanin G. Measurement invariance: Review of practice and implications. Hum Resour Manag Rev. 2008 Dec;18(4):210–22.

Reviewer #3: 

1. In page number 22, row number 456, table 5. Trimmed regression estimated predicting alcohol use frequency, the predictor Alcohol-involved is showing confidence intervel of 0.97-1.19, whereas the p value is significant. Could you explain this?

Response: Thank you for bringing this to our attention. The standardized beta value for the alcohol-involved predictor was a typo. The correct value has been entered into the table, and both the confidence interval and p-value indicate a significant finding. 

2. In the discussion section few sentences are very lengthy (row number 477-480), please make such sentences small for easy understanding by the readers.

Response: We have edited the discussion section to shorten sentences and improve clarity. 

3. I felt that limited references were used in the discussion sections (32 to 36) to strenghten your results. I would sugest to incorporate few more studies to improve the vigour of your paper. Please use the the following study (if apt)!

a) Love My Body: Pilot Study to Understand Reproductive Health Vulnerabilities in Adolescent Girls, doi:10.2196/16336

Response: We have edited the discussion section to provide more references to previous work. 

4. Please format all the references in simillar way, as reference 1 to 9 is arranged differently than the rest.

Response: We have re-formatted the references for consistency.

Please further strenghten your conclusion, as the study is providing some important outcomes, but the same is not getting reflected in your discussion and recommendations.

Response: We have added additional points to the discussion section as well as conclusion to highlight implications and novelty of the current study.

Reviewer #4: 

The manuscript is technically sound and all the data support the conclusions. All the statistical analysis been performed appropriately and rigorously. All the data is verified and free of plagiarism and written in an intelligible fashion and written in standard English.

Response: Thank you for taking the time to review our manuscript. 

Reviewer #5: 

I have few queries and suggestion for betterment of your manuscript.

1. why you just take additional data such as media use, dating, and time spent alone only? There are even more variables which may incompatible of alcohol use.

Response: We appreciate this question. Because we conducted secondary analysis of national survey data, we were limited to the available data that were collected by the national study. In this regard, we used all available data from the national study, and selected all possible variables that may facilitate or be incompatible with alcohol use. Although the literature has identified other variables that may be incompatible with alcohol use, the national study did not measure these variables, and thus, we were not able to include them in the current study. We have added this as a limitation to the discussion section.

2. In result section you have mentioned data if missing variable. how do you define it ?

Response: We added a statement to the methods section specifying how missing data were defined (p 15). Missing data were defined and identified by the national study in their codebook (Miech et al., 2019), and could occur if students were absent the during data collection or if students transferred to another school or dropped out from school during data collection. Additionally, missing data may have occurred if participants skipped a survey item at random.

Reference:

Miech RA, Johnston LD, Bachman JG, O’Malley PM, Schulenberg JE, Patrick ME. Monitoring the Future: A Continuing Study of American Youth (8th- and 10th-Grade Surveys), 2019: Version 1 [Internet]. ICPSR - Interuniversity Consortium for Political and Social Research; 2020 [cited 2023 Jan 23]. Available from: https://www.icpsr.umich.edu/web/NAHDAP/studies/37842/versions/V1

3. Along with behavioural economic theory, this manuscript result minghr also supported by integrated behavioural model. (Please refer it to.. )

Response: In the discussion section, we have included reference to the Integrated Behavioral Model, and recommend future research examine attitude towards these activities, social norms towards activities, and self-perceived control regarding these activities in order to better understand activity engagement and alcohol use among adolescents. 

5. If you make cut off and analysis om three categories of adolescent the result might look interesting; early adolescence, middle, and late adolescence

Response: We appreciate this suggestion; however, unfortunately we are unable to examine adolescents by sub-groups due to limitations in the available data from the national parent survey. The available data only included participants who were 12-14 years old (enrolled in 8th grade) and 15-17 years old (enrolled in 10th grade). We did assess for measurement invariance between the 8th and 10th grade samples to determine if we could compare the samples directly. However, findings indicated the survey questions performed differently between the two groups, and thus we could not make direct comparisons between age groups for reasons outlined in Little (2013) and Schmitt & Kuljanin (2008). Thus, we selected the 10th grade sample (~15-17 years old) as the focus of the study, given the higher drinking prevalence among the 15-17 years old/10th grade group. We had added information regarding this decision to the methods section. 

References:

Little TD. Longitudinal structural equation modeling. New York: The Guilford Press; 2013. 386 p. (Methodology in the social sciences).

Schmitt N, Kuljanin G. Measurement invariance: Review of practice and implications. Hum Resour Manag Rev. 2008 Dec;18(4):210–22.

---

## [Editor Report · Decision Letter 1]

25 Aug 2023

Engagement in types of activities and frequency of alcohol use in a national sample of United States adolescents

PONE-D-23-02338R1

Dear Dr. Fazzino,

We’re pleased to inform you that your manuscript has been judged scientifically suitable for publication and will be formally accepted for publication once it meets all outstanding technical requirements.

Kind regards,

Chiranjivi Adhikari, MPH, MHEd., PhD Candidate

Academic Editor

PLOS ONE
---

## [Editor Report · Acceptance letter]

1 Sep 2023

PONE-D-23-02338R1 

Engagement in types of activities and frequency of alcohol use in a national sample of United States adolescents 

Dear Dr. Fazzino:

I'm pleased to inform you that your manuscript has been deemed suitable for publication in PLOS ONE. Congratulations! Your manuscript is now with our production department. 

Kind regards, 

on behalf of

Mr. Chiranjivi Adhikari 

Academic Editor

PLOS ONE